# Low-Cost Exoskeletons for Learning Whole-Arm Manipulation in the Wild

**Abstract:** While humans can use parts of their arms other than the hands for manipulations like gathering and supporting, whether robots can effectively learn and perform the same type of operations remains relatively unexplored. As these manipulations require joint-level control to regulate the complete poses of the robots, we develop *AirExo*, a low-cost, adaptable, and portable dual-arm exoskeleton, for teleoperation and demonstration collection. As collecting teleoperated data is expensive and time-consuming, we further leverage *AirExo* to collect cheap in-the-wild demonstrations at scale. Under our in-the-wild learning framework, we show that with only 3 minutes of the teleoperated demonstrations, augmented by diverse and extensive in-the-wild data collected by *AirExo*, robots can learn a policy that is comparable to or even better than one learned from teleoperated demonstrations lasting over 20 minutes. Experiments demonstrate that our approach enables the model to learn a more general and robust policy across the various stages of the task, enhancing the success rates in task completion even with the presence of disturbances.

**Keywords:** AirExo, In-the-Wild Learning, Data Collection

## 1 Introduction

Robotic manipulation has emerged as a crucial field within the robot learning community and attracted significant attention from researchers. With the advancement of technologies such as deep learning, robotic manipulation has evolved beyond conventional grasping [9, 11, 33] and pick-and-place tasks [32, 43], encompassing a diverse array of complex and intricate operations [2, 3, 6, 10].

Most of the current robotic manipulation research focuses on interacting with the environment solely with the end-effectors of the robots, which correspond to the hands of human beings. However, as humans, we can also use other parts of our arms to accomplish or assist with various tasks in daily life. For example, holding objects with lower arms, closing fridge door with elbow, *etc*. In this paper, we aim to investigate and explore the ability of robots to effectively execute such tasks. To distinguish from the classical manipulation involving end-

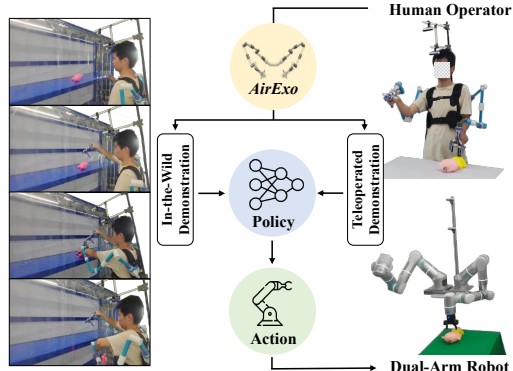

**Figure 1:** The methodology of our in-the-wild learning framework with low-cost exoskeletons *AirExo*. It empowers the human operator to not only control the dual-arm robots for collecting teleoperated demonstrations but also directly record in-the-wild demonstrations. Besides commonly-used teleoperated demonstrations, our learning framework also leverages the extensive and cheap in-the-wild demonstrations in policy learning, resulting in a more general and robust policy compared to training with even more teleoperated demonstrations.

effectors, we refer to these actions as **whole-arm manipulation**. Since most whole-arm manipulation tasks require the coordinated collaboration of both limbs, we formalize them into the framework of the bimanual manipulation problem.

While whole-arm manipulation is natural and simple for humans, it can become challenging for robots. First, whole-arm manipulation usually implies extensive contact with the surrounding environment and collision risks during manipulation. Second, whole-arm manipulation necessitates precise movement of the entire robot pose, as opposed to the conventional methods of only reaching the end-effector pose at the destination. An intuitive approach to address these two challenges is to adapt joint-level control for robots. To enable that, we adopt a joint-level imitation learning schema, wherein joint-level control is needed when collecting the robot demonstration.

Recently, Zhao *et al.* [46] introduced an open-source low-cost ALOHA system which exhibits the capability to perform joint-level imitation learning through real-world teleoperated data. ALOHA system leverages two small, simple and modular bimanual robots ViperX [37] and WidowX [40] that are almost identical to each other, to establish a leader-follower framework for teleoperation. Due to the limited payload of the robots, they focus more on fine-grained manipulation. Besides, their hardwares cannot be seamlessly adapted to other robots commonly employed for laboratory research or industrial purposes. Similarly, while several literatures [8, 15, 17, 19, 45] also designed special exoskeletons for certain humanoid robots or robot arms, the cross-robot transferability of their exoskeletons remain a challenge.

To address the above issues, we develop *AirExo*, an *open-source*, *low-cost*, *robust* and *portable* dual-arm exoskeleton system that can be quickly modified for different robots. All structural components of *AirExo* are *universal* across robots and can be fabricated entirely through 3D printing, enabling easy assembly even for non-experts. After calibration with a dual-arm robot, *AirExo* can achieve precise joint-level teleoperations of the robot.

Contributed to its portable property, *AirExo* enables *in-the-wild data collection for dexterous manipulation without needing a robot*. Humans can wear the dual-arm exoskeleton system, conduct manipulation in the wild, and collect demonstrations at scale. This breakthrough capability not only simplifies data collection but also extends the reach of whole-arm manipulation into unstructured environments, where robots can learn and adapt from human interactions. The one-to-one mapping of joint configurations also reduces the barriers of transferring policies trained on human-collected data to robots. Experiments show that with our in-the-wild learning framework, the policy can become more sample efficient for the expensive teleoperated demonstrations, and can acquire more high-level knowledge for task execution, resulting in a more general and robust strategy. The source code, data and exoskeleton models will be made publicly available.

## 2 Related Works

**Imitation Learning** Imitation learning has been widely applied in robot learning to teach robots how to perform various tasks by observing and imitating demonstrations from human experts. One of the simplest methods in imitation learning is behavioral cloning [27], which learns the policy directly in a supervised manner without considering intentions and outcomes. Most approaches parameterize the policy using neural networks [2, 5, 31, 44, 46], while non-parametric VINN [26] leverages the weighted $k$-nearest-neighbors algorithm based on the visual representations extracted by BYOL [14] to generate the action from the demonstration database. This simple but effective method can also be extended to other visual representations [22, 23, 25, 29] for robot learning. In the context of imitation learning for bimanual manipulation, Xie *et al.* [41] introduced a paradigm to decouple the high-level planning model into the elemental movement primitives. Several literature have focused on designing special frameworks to solve specific tasks, such as knot tying [18, 34], banana peeling [17], culinary activities [21], and fabric folding [39]. Addressing the challenge of non-Markovian behavior observed in demonstrations, Zhao *et al.* [46] utilized the notion of action chunking as a strategy to enhance overall performance.

**Teleoperation** Demonstration data play a significant role in robotic manipulation, particularly in the methods based on imitation learning. For the convenience of subsequent robot learning, these demonstration data are typically collected within the robot domain. A natural approach to gather such demonstrations is human teleoperation [24], where a human operator remotely controls the robot to execute various tasks. Teleoperation methods can be broadly categorized into two classes based on their control objectives: one aimed at manipulating the end-effectors of the robots [2, 7, 10, 16, 30, 44] and one focused on regulating the complete poses of the entire robots, such as exoskeletons [8, 15, 17, 35, 45] and a pair of leader-follower robots [46]. For whole-arm manipulation tasks, we need to control the full pose of the robots, which makes exoskeletons a relatively favorable option under this circumstance.

**Learning Manipulation in the Wild** Despite the aforementioned teleoperation methods allow us to collect robotic manipulation data, the robot system is usually expensive and not portable, posing challenges to collect demonstration data at scale. To address this issue, previous research has explored the feasibility of learning from interactive human demonstrations, *i.e.* in-the-wild learning for robotic manipulation [1, 4, 19, 28, 33, 42]. In contrast to the costly robot demonstrations, in-the-wild demonstrations are typically cheap and easy to obtain, allowing us to collect a large volume of such demonstrations conveniently. Typically, there are two primary domain gaps for learning manipulation in the wild: (1) the gap between human-operated images and robot-operated images, and (2) the gap between human kinematics and robot kinematics. The former gap can be solved through several approaches: by utilizing specialized end-effectors that match the end-effectors of the robots [19, 42]; by initially pre-training with in-the-wild data and subsequently fine-tuning with robot data [33]; or by applying special image processing technique to generate agent-agnostic images [1]. The latter gap is currently addressed by applying structure from motion algorithms [33, 42], adopting a motion tracking system [28], or training a pose detector [1, 38] to extract the desired poses. However, these methods are not suitable for whole-arm dexterous manipulation, since motion tracking usually focuses on the end-effector, and pose detector is vulnerable to visual occlusions and does not map to the robot kinematics.

# 3 AirExo: An Open-Source, Portable, Adaptable, Inexpensive and Robust Exoskeleton

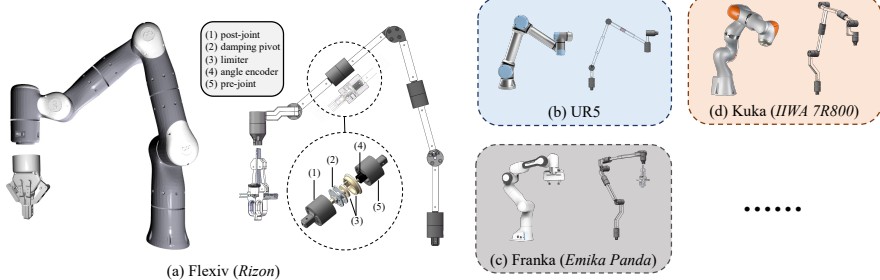

**Figure 2:** *AirExo* models for different types of robots. Notice that the internal structure of the joints is standardized, only the linkages are altered to accommodate different robotic arm configurations.

## 3.1 Exoskeleton

From the preceding discussions in Sec. 1, we summarize the following 5 key design objectives of an exoskeleton: (1) affordability; (2) adaptability; (3) portability; (4) robustness and (5) maintenance simplicity. Based on these design objectives, we develop *AirExo* as follows.

In this paper, we employ two Flexiv Rizon arms [12] for experiments. As a result, the structural design of *AirExo* is predominantly tailored to their specifications. Meanwhile, to ensure its universality, it can be easily modified for use with other robotic arms like UR5 [36], Franka [13] and

Kuka [20], as depicted in Fig. 2. Based on the morphology of our robot system, *AirExo* is composed of two symmetrical arms, wherein the initial 7 degree-of-freedoms (DoFs) of each arm correspond to the DoFs of the robotic arm, and the last DoF corresponds to the end-effector of the robotic arm. Here, we design a two-finger gripper with 1 DoF as an optional end-effector for each arm. Overall, *AirExo* is capable of simulating the kinematics of the robot across its entire workspace, as well as emulating the opening and closing actions of the end-effectors.

According to design objective (3), to improve the wearable experience for operators and concurrently enhance task execution efficiency, we dimension *AirExo* to be 80% of the robot's size, based on the length of the human arm. In the end-effector of the exoskeleton, we design a handle and a scissor-like opening-closing mechanism to simulate the function of a two-fingered gripper, while also facilitating gripping actions by the operator. The two arms of the exoskeleton are affixed to a base, which is mounted on a vest. This allows the operator to wear it stably, and evenly distributing the weight of the exoskeleton across the back of the operator to reduce the load on the arms, thereby enabling more flexible arm motions. Additionally, an adjustable camera mount can be installed on the base for image data collection during operations.

The joints of *AirExo* adapt a dual-layer structure, with the outer case divided into two parts: the portion proximate to the base is referred to as the *pre-joint*, while the other half is called the *post-joint*. As illustrated in Fig. 2(a), these two components are connected via a metal *damping pivot*, and their outer sides are directly linked to the connecting rod. *AirExo* primarily achieves high-precision and low-latency motion capture through the *angle encoders* (with a resolution of 0.08 degrees), whose bases are affixed to the *pre-joints*. The pivots of the encoders are connected to the *post-joint* through a *limiter*, which is comprised of a dual-layer disc and several steel balls to set the angle limit for each joint. The dual-layer joint structure ensures that the encoders remain unaffected by bending moments during motions, rotating synchronously with the joints, which safeguards the encoders and reduces failures effectively. This aligns with the design objective (4) and (5).

Except the fasteners, damping pivots, and electronic components, all other components of *AirExo* are fabricated using PLA plastic through 3D printing. The material has a high strength and a low density, thereby achieving a lightweight but robust exoskeleton. The prevalence of 3D-printed components allows the exoskeleton to be easily adapted to different robots. This adaptation entails adjusting the dimensions of certain components based on the target robot's specifications and subsequently reprinting and installing them, without modifying the internal structure. *AirExo* costs approximately $600 in total, which is in accordance with the design objective (1).

### 3.2 Calibration and Teleoperation

Since *AirExo* shares the same morphology with the dual-arm robot except for the scale, the calibration process can be performed in a quite straightforward manner. After positioning the robot arms at a specific location like a fully extended position, and aligning the exoskeleton to match the robot posture, we can record the joint positions $\{q_i^{(c)}\}_{i=1}^d$ and the encoder readings $\{p_i^{(c)}\}_{i=1}^d$ of *AirExo*, where $d$ denotes the DoFs. Consequently, during teleoperation, we only need to fetch the encoder readings $\{p_i\}_{i=1}^d$ and transform them into the corresponding joint positions $\{q_i\}_{i=1}^d$ using Eqn. (1), and let the robot moves to the desired joint positions:

$$q_i = \min\left(\max\left(q_i^{(c)} + k_i(p_i - p_i^{(c)}), q_i^{\min}\right), q_i^{\max}\right), \tag{1}$$

where $k_i \in \mathbb{R}$ is the coefficient controlling direction and scale, and $q_i^{\min}, q_i^{\max}$ denote the joint angle limits of the robotic arms. Typically, we set $k = \pm 1$, representing the consistency between the encoder direction of the exoskeleton and the joint direction of the robot. For grippers, we can directly map the angle range of the encoders to the opening and closing range of the grippers for teleoperation.

After calibration, the majority of angles within the valid range of the robot arms can be covered by the exoskeleton. Given that the workspaces of most tasks fall within this coverage range, we can teleoperate the robot using the exoskeleton conveniently and intuitively. If a special task $t$ needs a

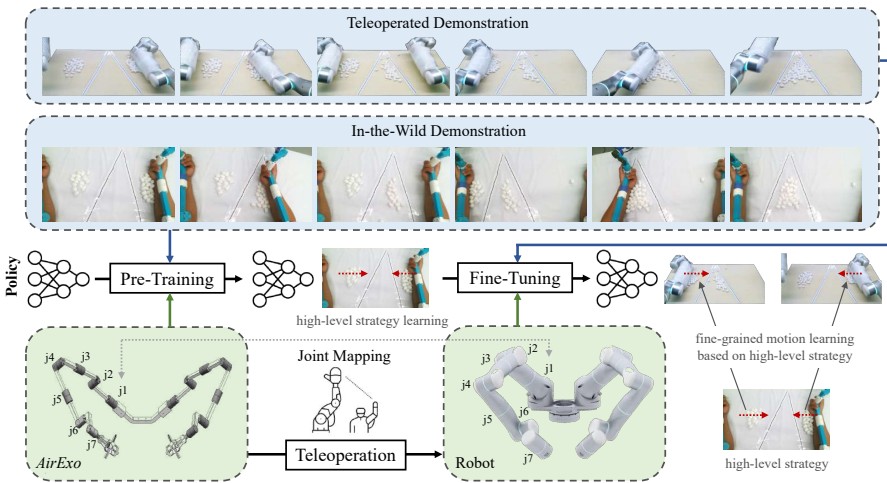

**Figure 3:** Overview of learning whole-arm manipulations in the wild with *AirExo*. First, we use in-the-wild demonstrations and exoskeleton actions that are transformed into the robot's domain to pre-train the policy, which corresponds to learning the high-level strategy of task execution. Then, we use teleoperated demonstrations and robot actions to fine-tune the policy, which corresponds to learning fine-grained motion based on the learned high-level strategy.

wider operation range, we can simply scale the exoskeleton range using coefficients $k_i$, and apply task-specific joint constraint $[q_i^{t,\min}, q_i^{t,\max}]$ instead of original kinematic constraint in Eqn. (1) for better teleoperation performance.

## 3.3 In-the-Wild Learning with AirExo

For in-the-wild whole-arm manipulation learning, we install a camera (or cameras under multi-camera settings) on the camera mount of *AirExo* in roughly the same position(s) as the camera(s) on the robot. Using this configuration, images from both teleoperated demonstrations and in-the-wild demonstrations exhibit a relatively similar structure, which is advantageous for policy learning.

Our approach to learn whole-arm manipulation in the wild with *AirExo* is illustrated in Fig. 3. As we discussed in Sec. 2, *AirExo* serves as a natural bridge for the kinematic gap between humans and robots. To address the domain gap between images, our approach involves a two-stage training process. In the first stage, we pre-train the policy using in-the-wild human demonstrations and actions recorded by the exoskeleton encoders. During this phase, the policy primarily learns the high-level task execution strategy from the large-scale and diverse in-the-wild human demonstrations. Subsequently, in the second stage, the policy undergoes fine-tuning using teleoperated demonstrations with robot actions to refine the motions based on the previously acquired high-level task execution strategy.

As previously discussed in Sec. 3.1, we resize the exoskeleton to ensure its wearability. Some concerns may arise regarding whether this scaling adjustment could impact the policy learning process. Here, we argue that it has a minimal effect on our learning procedure. Firstly, the core kinematic structure, essential for our learning framework, remain unaffected by the resizing. Thus human demonstrations preserve the fundamental dynamics of the system. Secondly, our approach does not impose strict alignment requirements between human demonstration images and robot images. We find that similar visual-action pairs collected by our exoskeleton effectively support the pretraining stage, without demanding precise visual matching between human and robot demonstrations.

We use the state-of-the-art bimanual imitation learning method ACT [46] for policy learning. Our experiments demonstrate that it can indeed learn the high-level strategy through the pre-training process and significantly enhance the evaluation performance of the robot and the sample efficiency of the expensive teleoperated demonstrations.

## 4 Experiments

In this section, we conduct experiments on 2 whole-arm tasks to evaluate the performance of the proposed learning method. All demonstration data are collected by *AirExo*.

### 4.1 *Gather Balls*: Setup

**Task**  Two clusters of cotton balls are randomly placed on both sides of the tabletop (40 balls per cluster). The goal is to gather these balls into the designated central triangular area using both arms. The process of this contact-rich task is illustrated in Fig. 4.

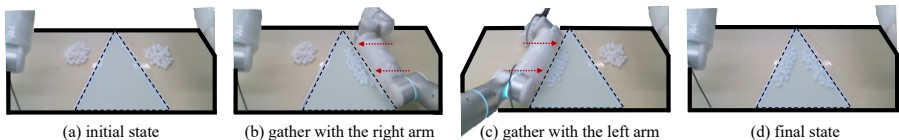

(a) initial state    (b) gather with the right arm    (c) gather with the left arm    (d) final state

**Figure 4:** Definition of *Gather Balls* task. The goal is to gather the balls into the central triangular area, which is highlighted in light blue. The red dashed arrows denote the motions of the robot arms. Sponge paddings are used to envelop the external surface of the robot arms to diminish the mechanical failures arising from contacts.

**Metrics**  We consider the percentage of balls being allocated within the central triangular area as the task completion rate $c$ (if a ball is precisely on the line, it is considered a half), including both the completion rates of the left arm and the right arm. Simultaneously, task success is defined as the task completion rate exceeding a certain threshold $\delta$. In this experiment, we set $\delta = 40\%, 60\%, 80\%$. We also record the collision rate to gauge the precision of the operations.

**Methods**  We employ VINN [26] and its variants that alter the visual representations [22, 23, 29] as non-parametric methods. Other methods include ConvMLP [44], BeT [31] and ACT [46]. All of them are designed for joint-space control or can be easily adapted for joint-space control. We apply our proposed learning approach to ACT for learning from in-the-wild demonstrations. For all methods, we carefully select the hyper-parameters to ensure better performance.

**Protocols**  The evaluation is conducted on a workstation equipped with an Intel Core i9-10980XE CPU. The time limit is set as 60 seconds per trial. Given that all methods can operate at approximately 5Hz, resulting in a total of 300 steps for the evaluation, the time constraint proves sufficient for the task. We conduct 50 consecutive trials to ensure stable and accurate results, calculating the aforementioned metrics.

### 4.2 *Gather Balls*: Results and Analyses

The experimental results on the *Gather Balls* task are shown in Tab. 1. When using 50 teleoperated demonstrations as training data, VINN performs the best among all non-parametric methods, while ACT excels among all parametric methods. When using only 10 teleoperated demonstrations for training, the performance of both VINN and ACT degrades inevitably. However, after applying our in-the-wild learning framework, with the assistance of in-the-wild demonstrations, ACT can achieve the same level of performance as 50 teleoperated demonstrations with just 10 teleoperated demonstrations. This demonstrates that our learning framework with in-the-wild demonstrations makes the policy more sample-efficient for teleoperated demonstrations.

We then delve into the experimental results to provide more insights about why and how our learning framework works. When analyzing the failure cases of different methods in the experiments in Fig. 5(a), we find that the ACT policy trained solely on teleoperated demonstrations exhibits an issue of imbalance between accuracies of two arms, with better learning outcomes for the left arm. This imbalance becomes more pronounced as the number of teleoperated demonstrations decreases to 10. With the help of the in-the-wild learning stage, the policy becomes more balanced between

| # Demos | | Method | Completion Rate $c$ (%) ↑ | | | Success Rate (%) ↑ | | |
|---|---|---|---|---|---|---|---|---|
| Teleop. | i.t.w | | Overall | Left | Right | $c \geq 80$ | $c \geq 60$ | $c \geq 40$ |
| 50 | - | VIP [22] + NN | 27.74 | 0.02 | 55.45 | 0 | 0 | 36 |
| 50 | - | VC-1 [23] + NN | 52.54 | 32.53 | 72.55 | 4 | 42 | 74 |
| 50 | - | MVP [29] + NN | 55.10 | 58.55 | 62.00 | 12 | 62 | 76 |
| 50 | - | VINN [26] | **76.88** | 75.73 | 78.03 | **58** | **84** | 94 |
| 50 | - | ConvMLP [44] | 15.56 | 2.35 | 28.78 | 0 | 0 | 2 |
| 50 | - | BeT [31] | 24.66 | 7.38 | 41.95 | 0 | 2 | 32 |
| 50 | - | ACT [46] | 75.61 | 94.63 | 56.60 | 54 | 70 | **100** |
| 10 | - | VINN [26] | 68.68 | 60.28 | 77.08 | 36 | 76 | 88 |
| 10 | - | ACT [46] | 64.31 | 91.95 | 36.68 | 24 | 60 | **96** |
| 10 | 50 | ACT [46] | 73.76 | 88.83 | 58.70 | **62** | 72 | 88 |
| 10 | 100 | ACT [46] | **75.15** | 75.63 | 74.68 | 56 | **80** | 88 |

**Table 1:** Experimental results on the *Gather Balls* task. Here "teleop." denotes teleoperated demonstrations and "i.t.w." denotes in-the-wild demonstrations.

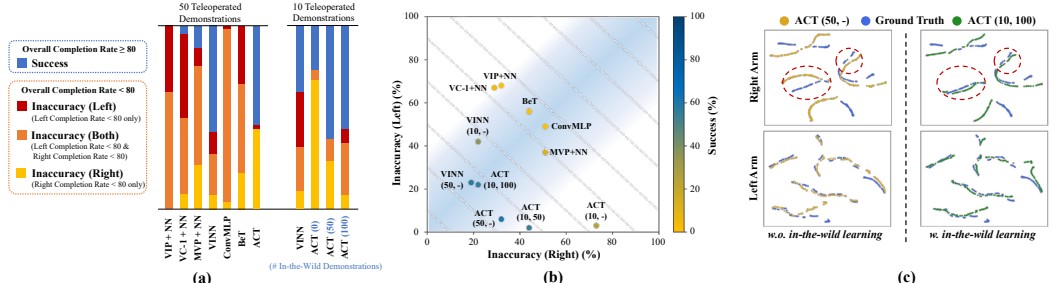

**Figure 5:** Analyses of methods on the *Gather Balls* task. Here we define the overall completion rate over 80% as success. **(a)** We analyze the failure causes of each method in every trial. **(b)** We amortize the inaccuracy (both) rate evenly into the inaccuracy (left) and inaccuracy (right) rates, and draw a comparison plot of failure modes for different methods. $(x, y)$ means the policy is trained with $y$ in-the-wild demonstrations then $x$ teleoperated demonstrations. The dashed lines represent contour lines with the same success rate, and the regions with light blue background imply a more balanced policy between left and right arms. **(c)** $t$-SNE visualizations of the ground-truth actions and the policy actions w/wo in-the-wild learning on the validation set.

two arms even with fewer teleoperated demonstrations, as shown in Fig. 5(b). From Fig. 5(c), we also observe that the policy focuses more on learning the motions of the right arm when cooperated with in-the-wild learning, as highlighted in red dashed circles, while keeping the accurate action predictions on the left arm. We believe that this is attributed to the extensive, diverse, and accurate in-the-wild demonstrations provided by *AirExo*, enabling the policy to acquire high-level strategy knowledge during the pre-training stage. Consequently, in the following fine-tuning stage, it can refine its actions based on the strategy, thus avoiding learning actions blindly from scratch.

### 4.3 *Grasp from the Curtained Shelf*: Setup and Results

**Task**   A cotton toy is randomly placed in the center of a shelf with curtains. The goal is to grasp the toy and throw it into a bin. To achieve it, the robot needs to use its right arm to push aside the transparent curtain first, and maintain this pose during the following operations. The process of this multi-stage task is illustrated in Fig. 6.

**Metrics, Methods, and Protocols**   We calculate the average success rate at the end of each stage as metrics. Based on the experimental results on the *Gather Balls* task, we select VINN [26] and ACT [46] as methods in experiments, as well as ACT equipped with our in-the-wild learning framework. The evaluation protocols are the same as the *Gather Balls* task, except that the time limit is 120 seconds (about 400 steps) and the number of trials is 25.

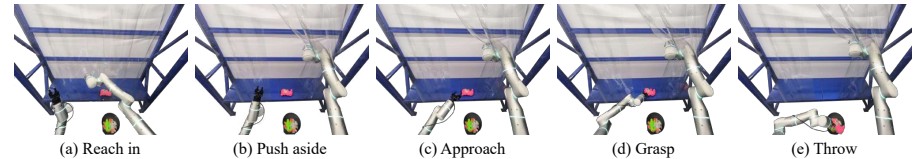

| (a) Reach in | (b) Push aside | (c) Approach | (d) Grasp | (e) Throw |

**Figure 6:** Definition of the *Grasp from the Curtained Shelf* task. The robot needs to (a) reach in its right arm to the transparent curtain and (b) push aside the curtain, then (c) approach the object with its left arm, (d) grasp the object and finally (e) throw the object.

**Results** The results are given in Tab. 2. Similar to the results of the *Gather Balls* task, as the number of training teleoperated demonstrations is reduced, both VINN and ACT experience a decrease in success rates, especially in the later "throw" stage. However, after training with our

| # Demos | | Method | Success Rate (%) ↑ | | | | |
|---|---|---|---|---|---|---|---|
| Teleop. | i.t.w. | | Reach in | Push aside | Approach | Grasp | Throw |
| 50 | - | VINN [26] | **100** | 96 | 92 | 60 | 48 |
| 50 | - | ACT [46] | **100** | **100** | **100** | **84** | **84** |
| 10 | - | VINN [26] | **100** | 84 | 84 | 60 | 44 |
| 10 | - | ACT [46] | **100** | **100** | 96 | 72 | 44 |
| 10 | 50 | ACT [46] | **100** | **100** | 96 | 76 | 76 |
| 10 | 100 | ACT [46] | **100** | **100** | **100** | **92** | **88** |

**Table 2:** Experimental results on the *Grasp from the Curtained Shelf* task.

in-the-wild learning framework, ACT exhibits a significant improvement in success rates in the "grasp" and "throw" stages. It achieves even higher success rates, surpassing those obtained with the original set of 50 teleoperated demonstrations lasting more than 20 minutes, using only 10 such demonstrations lasting approximately 3 minutes. This highlights that our proposed in-the-wild framework indeed enables the policy to learn a better strategy, effectively enhancing the success rates in the later stages of multi-stage tasks.

**Robustness Analysis** We design three kinds of disturbances in the robustness experiments to explore whether in-the-wild learning improves the robustness of the policy. The results shown in Tab. 3 demonstrate that our in-the-wild learning framework can leverage diverse in-the-wild demonstrations to make the learned policy more robust and generalizable to various environmental disturbances.

| Disturbances | w/wo i.t.w. learning | Success / All |
|---|---|---|
| Novel Object | ✗ | 4 / 8 |
| | ✔ | **7** / 8 |
| Different Background | ✗ | 2 / 8 |
| | ✔ | **6** / 8 |
| Visual Distractors | ✗ | 4 / 8 |
| | ✔ | **8** / 8 |

**Table 3:** Results of the robustness experiments on the *Grasp from the Curtained Shelf* task.

## 5  Conclusion

In this paper, we develop *AirExo*, an open-source, low-cost, universal, portable, and robust exoskeleton, for both joint-level teleoperation of the dual-arm robot and learning whole-arm manipulations in the wild. Our proposed in-the-wild learning framework decreases the demand for the resource-intensive teleoperated demonstrations. Experimental results show that policies learned through this approach gain a high-level understanding of task execution, leading to improved performance in multi-stage whole-arm manipulation tasks. This outperforms policies trained from scratch using even more teleoperated demonstrations. Furthermore, policies trained in this framework exhibit increased robustness in the presence of various disturbances.

In the future, we are excited to see our *AirExo* collecting large-scale demonstrations in unstructured environments and facilitating robot learning. We will investigate how to better address the image gap between in-the-wild data in the human domain and teleoperated data in the robot domain, enabling robots to learn solely through large-scale in-the-wild demonstrations with *AirExo*, thus further reducing the learning cost.

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
