# OpenReview forum: "Low-Cost Exoskeletons for Learning Whole-Arm Manipulation in the Wild"
_robot-learning.org/CoRL/2023/Workshop/TGR — CoRL 2023 Workshop TGR Oral_

### Official Review · Reviewer_jN46 · 2023-10-16

**Rating:** 7
**Confidence:** 3

**Review:**

This paper presents an exoskeleton for collecting human's demonstration for whole-arm manipulation task. The proposed system could help generalist robots to learn more diverse, robust, and sophisticated skills.

---

### Official Review · Reviewer_sjJp · 2023-10-19

**Rating:** 8
**Confidence:** 3

**Review:**

This paper develop AirExo, a low-cost, adaptable, and portable dual-arm exoskeleton, for joint-level teleoperation and demonstration collection. Beyond end effector (or analogously, hands of human), their system allows to explore the possibility of whole-arm manipulation, which uses other parts of the arms to accomplish or assist with various tasks in daily life similar to human. They demonstrate an interesting route toward a more general manipulation setting as well as a relatively more scalable approach to collect human demonstration in the real world compared to teleoperation with other types of human interfaces like control stick. This work may stimulate interesting discussion in the workshop.

---

### Decision · Program_Chairs · 2023-10-20

**Decision:**

Accept (Oral)

**Comment:**

Very cool device for scaling up real-world data collection!